# An analysis of government-sponsored health insurance enrolment and claims data from Meghalaya: Insights into the provision of health care in North East India

Eliza K. Dutta[1]*, Sampath Kumar[2], Selvaraju Venkatachalam[3], Laura E. Downey[4,5‡], Sandra Albert[1‡]

1 Indian Institute of Public Health Shillong, Shillong, Meghalaya, India, 2 Department of Health & Family Welfare, Government of Meghalaya, Shillong, India, 3 Consultant (Health Economist), New Delhi, India, 4 The George Institute for Global Health, University of New South Wales, Sydney, Australia, 5 School of Public Health, Imperial College London, London, United Kingdom

‡ LED and SA are joint senior authors on this work.
* eliza.dutta@iiphs.org

**Data Availability Statement:** The data are owned by the Megha-Health Insurance Scheme, Directorate of Health Services, Government of

## Abstract

### Introduction

The Megha Health Insurance Scheme (MHIS) was launched in 2013 in the North-East Indian state of Meghalaya to reduce household out-of-pocket expenditure on health and provide access to high-quality essential healthcare. Despite substantial expansion of the MHIS since the scheme's inception, there is a lack of comprehensive documentation and evaluation of the scheme's performance against its Universal Health Care (UHC) objectives.

### Methods

We analysed six years of enrolment and claims data (2013–2018) covering three phases of the scheme to understand the pattern of enrolment, utilisation and care provision under the MHIS during this period. De-identified data files included information on age, sex, district of residence, the district of provider hospital, type of hospital, date of admission, status at discharge, claimed category of care, package codes, and amount claimed. Descriptive statistics were generated to investigate key trends in enrolment, service utilisation, and Government health spending under the MHIS.

### Results

Approximately 55% of the eligible population are currently enrolled in MHIS. Enrolment increased consistently from phase I through III and remained broadly stable across districts, gender, age group and occupation categories, with a small decline in males 19–60 years. Claims were disproportionately skewed towards private provision; 57% of all claims accrued to the 18 empanelled private hospitals and 39% to the 159 public sector facilities. The package 'General Ward Unspecified' was responsible for the highest volume of claims and highest financial dispensation across all three phases of the scheme. This likely indicates substantial administrative error and is potentially masking both true burden of disease and

Meghalaya. The data was provided to the Indian Institute of Public Health Shillong as it is mandated to carry out studies on economic evaluation for the states in the north-east region of India. Aggregated data are already available for public consumption, can be accessed at (https://mhis.org.in/mhis-i-enrolment-district-blocks/). Interested researchers can also write directly to the MHIS, Directorate of Health Services, Government of Meghalaya and request the necessary data with appropriate justification for their research work. The appropriate contacts can be accessed through the following link: (https://mhis.org.in/contacts/). This link contains contact of the key information person for each district. This page also has the helpdesk phone numbers for additional queries. This should provide all key contacts and necessary information, should any future researchers wish to directly contact relevant departments of the Megha Health Insurance Scheme to undertake further research. Interested researchers can also write directly to the MHIS, Directorate of Health Services, Government of Meghalaya and request the necessary data with appropriate justification for their research work. The appropriate contacts can be accessed through the following link: https://mhis.org.in/contacts/ This link contains contact of the key information person for each district. This page also has the helpdesk phone numbers for additional queries. This should provide all key contacts and necessary information, should any future researchers wish to directly contact relevant departments of the Megha Health Insurance Scheme to undertake further research.

**Funding:** We have received a grant, title 'Regional Resource Hub for Health Technology Assessment in the North Eastern Region (RRH-HTA-In)' from the Department of Health Research, Government of India, to conduct health economics projects. Grant no: F.NO.S.11011/02/2017-HR. This grant covers the salary of one of our study authors (ED). The Government of India has had no influence on the conduct, findings, or reporting of our study and remains an impartial funder. The funders had no role in the study design, data collection and analysis, decision to publish or preparation of the manuscript.

**Competing interests:** The authors have declared that no competing interests exist.

accurate financial provision for care under the MHIS. Anti-rabies injections for dog/cat bite contributed to 11% of total claims under MHIS III, and 1.6% of all claims under MHIS II. This warrants investigation to better understand the burden of animal bites on the Meghalayan population and inform the implementation of cost-effective strategies to reduce this burden.

## Conclusions

This paper describes the first analysis of health insurance enrolment and claims data in the state of Meghalaya. The analysis has generated an important evidence base to inform future MHIS enrolment and care provision policies as the scheme expands to provide Universal Health Coverage to the state's entire population.

## Introduction

India is home to approximately one sixth of the global population and has experienced rapid economic growth over the last decade [1]. Improvements in population health over this period have been less marked, with the largest number of cases of tuberculosis globally, and a growing burden of chronic non-communicable diseases, accompanied by high rates of child malnutrition, stunting and maternal and neonatal mortality across many states [2]. The Government of India presently spends roughly 1.3% of GDP on healthcare [3]—one of the lowest public expenditures on health in the world. Despite this, the Government is committed to achieving Universal Health Coverage (UHC) for the Indian population [4]. This means that every rupee spent on health must be spent wisely to maximise population health gains within the confines of limited resources.

Health is constitutionally a state responsibility within India's federal structure; however, the central Government plays a major role in designing and implementing health programs and services in partnership with state Governments through the public healthcare network [5]. Both the central and various state Governments have experimented with a range of strategies to scale-up resources available for essential healthcare services, improve the accessibility and quality of care provided, and afford the poorest households with financial risk protection from catastrophic health expenditure [6].

In a country as vast as India, the success of Government-initiated health reforms in terms of expanding population coverage of essential health services, and reducing catastrophic out of pocket expenditure has varied widely. While some states have observed moderate improvements in healthcare access and quality with the implementation of progressive health reforms, such as the Aarogyasri social health insurance in Andhra Pradesh [7], or the similar Bhamashah Swasthya Bima Yojana (BSBY) in Rajasthan [8]. others have historically demonstrated significant difficulty in absorbing and diffusing the already limited Government health funding allocated towards improvements in health care delivery [9]. Launched in 2008, Rashtriya Swasthya Bima Yojana (RSBY) was India's first centrally-administered national health insurance programme for the Indian poor [10]. The scheme aimed to provide health insurance coverage to those classified as working within the 'unorganised' sector and living below the poverty line. This scheme was succeeded in 2018 with the much-anticipated Ayushman Bharat Pradhan Mantri Jan Arogya Yojana (AB-PMJAY), which expanded both the range of services provided and the pool of potential beneficiaries, and is intended to provide free access to healthcare for some 500 million people in the country [11]. The scheme was designed through a consultative process between central and state Governments, and implemented by the

National Health Authority and nodal State Health Authorities, though states have the flexibility to modify terms of beneficiary eligibility and the package of services available to suit local needs.

The state of Meghalaya is situated in the north east of India, with a population of 3 million. (12) Approximately 20 percent of the Meghalayan population live in urban dwellings, with the remaining 80 percent residing in rural communities [12]. The Government of Meghalaya was the first in India to promise UHC to its population regardless of their socio-economic status via the scale-up and expansion of its local health insurance scheme–the Megha Health Insurance Scheme (MHIS) [13]. The MHIS was first launched in 2012 and provides access to secondary and tertiary care to all enrolled citizens of the state through a network of empanelled public and private hospitals. It was subsumed within the auspices of the AB-PMJAY in 2018, where service coverage is provided according to a list of included packages of care defined by the National Health Authority (NHA) and continues to be implemented by the MHIS office, who operate as the administering State Health Authority (SHA). The scheme offers financial protection in case of hospitalization due to medical emergencies and covers a wide range of health conditions and services free at the point of delivery. Most care-packages include secondary and tertiary care, and pharmaceuticals are provided for, if administered directly to patients in empanelled hospitals.

As the Government of Meghalaya looks to refine and further expand the MHIS under their ambitious goal to provide the entire state population with UHC [13], it is important to first assess the operational performance of the scheme to date in order to create a robust evidence base from which informed decisions for shaping the future of the MHIS can be made. The objectives of this analysis were threefold: to understand the success of the MHIS in reaching target beneficiaries for enrolment; to identify the patterns of care provision and key trends in service utilisation and spending; and to generate evidence-based recommendations for consideration by the government to enhance the operational performance of the MHIS.

## Methods

A retrospective descriptive analysis was undertaken to identify patterns of enrolment, service provision and health spending in Meghalaya over the past six years (2013–2018).

### Data

De-identified medical insurance enrolment and claims data were made available to the research team at the Indian Institute of Public Health Shillong (IIPHS). Data comprised three iterations of the MHIS scheme, referred to herein as MHIS I (May 2013—July 2015), MHIS II (August 2015 –December 2016), and MHIS III (January 2017 –September 2018). All patient identifiable information was removed by the MHIS administrators prior to data transfer. A total of 43 Microsoft excel files were transferred by the MHIS team. The enrolment files included one line of data for each unique enrolee with information on date of enrolment, age, gender, and region of residence. The claims files, which could be linked to the enrolment files by a unique registration number, had one line of data for each procedure claimed for at each visit. Each line of claims data also included information on where the care was provided, which type of facility provided the care, the amount claimed, the length of stay associated with the claim, and the package of care under which the procedure was claimed for.

### Data analysis

Enrolment and claims data files were transferred into STATA (STATA v 14.2) for data cleaning and analysis. These files were manually checked against their respective data dictionaries

for values that were missing or incomplete. A high level of heterogeneity in values entered under 'package claimed' was identified, necessitating the research team to examine each line in the claims data files manually in order to identify aberrant package information and match it to the appropriate package text, where possible. Common reasons for heterogeneity included typing and/or spelling mistakes, and additional or missing spaces between words. The MHIS team was consulted to resolve queries and data discrepancies. Where it was unclear what package was claimed for and this could not be resolved, these lines were removed from the analyses.

Enrolment data was analysed by distribution across locations (districts), socio-economic status (households below and above poverty line), and by demographic variables (age and sex). Claims data were analysed by volume of individual claims and total financial provision for aggregate individual claims under each package. Descriptive statistics were generated for both enrolment and claims data, disaggregated across the three time periods of interest (MHIS I (May 2013—July 2015), MHIS II (August 2015 –December 2016), and MHIS III (January 2017 –September 2018), and by geographic region, gender, and age. As the entire population of Meghalaya is eligible for coverage under the MHIS, except for those employed by the Government and their families, who are covered under the Employees State Insurance Scheme (ESIS), total eligible population was calculated by subtracting the number of households who identify as being covered by the ESIS from the state population. The number of MHIS enrollees was then divided by the total eligible population in order to generate information on proportion of eligible population currently enrolled in the scheme.

## Results

### Enrolment data

Enrolment data is presented in Table 1.

Enrolment statistics indicated that a significant increase in enrolments was observed for MHIS II in comparison to MHIS I, with the number of enrollees almost doubling in the scheme's second iteration (MHIS I = 7,28,028 enrollees; MHIS II = 15,48,617). However, this increase was not observed for MHIS III, where the number of people enrolled in the scheme increased by less than 10,000 additional enrollees (15,57,008 enrolled). The number of current enrollees in MHIS as a proportion of the total number of eligible enrollees in the state obtained from census data indicate that 54.77% of the eligible population were enrolled in MHIS III.

### Claims data

**Care provider.** A breakdown of total claims made and total financial provision for claims by type of care provider is provided in Table 2.

Analysis of the most recent claims data under MHIS III (claims made between 2017–2018) revealed that more than half of the total amount claimed during this period (57%, INR 538,592,642) accrued to the 18 private hospitals empanelled under the scheme. The large network of public healthcare providers (159 in total) accounted for 39% of claims made (INR 367,048,292) during this period. In the earlier phases of MHIS I and II, the total amount of claims accrued to the private sector as a proportion of total amount claimed were 58% (MHIS II; INR 95,757,996) and 48% (MHIS I; INR 250,450,988) respectively. Financial provision for claims made outside of state increased from a 1% share of total spending to a 4% share of total spending between MHIS I and MHIS III.

**Volume of claims.** The top 15 packages as indicated by volume of claims in MHIS III are presented in Fig 1.

The highest package by volume was General Ward Unspecified (GWU) 42%, followed by cat/dog bite (11%), maternal care packages (normal delivery, antenatal care, vaginal delivery

**Table 1. Enrolment across all 3 phases of MHIS, 2013–2018.**

| Enrolment | MHIS I | MHIS II | MHIS III |
|---|---|---|---|
| **Age groups** | N (%) | N (%) | N (%) |
| 1–5 years | 97289 (13.4) | 157683 (10.2) | 200661 (12.9) |
| 6–18 years | 226729 (31.1) | 287985 (18.6) | 407208 (26.2) |
| 19–45 years | 295753 (40.6) | 832761 (53.8) | 712601 (45.8) |
| 46–60 years | 77707 (10.7) | 189832 (12.3) | 167103 (10.7) |
| >60 years | 30550 (4.2) | 80356 (5.2) | 69435 (4.5) |
| **Sex** | | | |
| Males | 325336 (44.7) | 825975 (53.3) | 742195 (47.7) |
| Females | 401724 (55.2) | 718494 (46.4) | 810214 (52.0) |
| Others | 968 (0.1) | 4148 (0.3) | 4599 (0.3) |
| **Districts** | | | |
| East Khasi Hills | 187883 (25.8) | 336800 (21.7) | 361167 (23.2) |
| West Khasi Hills | 62635 (8.6) | 130847 (8.4) | 146449 (9.4) |
| South West Khasi Hills | 31205 (4.3) | 56589 (3.7) | 58198 (3.7) |
| Ri Bhoi | 77928 (10.7) | 104736 (6.8) | 141774 (9.1) |
| East Jaintia Hills | 45982 (6.3) | 94804 (6.1) | 86709 (5.6) |
| West Jaintia Hills | 92450 (12.7) | 199027 (12.9) | 199426 (12.8) |
| West Garo Hills | 59121 (8.1) | 238392 (15.4) | 201128 (12.9) |
| South West Garo Hills | 72545 (10.0) | 143339 (9.3) | 129226 (8.3) |
| North Garo Hills | 48194 (6.6) | 103415 (6.7) | 93869 (6.0) |
| East Garo Hills | 28090 (3.9) | 60747 (3.9) | 71912 (4.6) |
| South Garo Hills | 21995 (3.0) | 79921 (5.2) | 67150 (4.3) |
| **Total** | **7,28,028** | **15,48,617** | **15,57,008** |

with episiotomy and peritoneal repair, caesarean delivery; 21.0% collectively), cataract care (1%), Intensive Care Unit (ICU) care (1%), and renal dialysis (0.9%). Data for MHIS II is presented in Fig 2 below.

In MHIS II, GWU represented over half of all claims by volume (59%), followed by maternal care (normal deliveries and peritoneum repair, caesarean delivery including lower segment C-section; 13.6%) and malaria (3%). Cat/dog bites, cataract care and ICU care contributed to 1.6%, 1.0% and 1.3% respectively in MHIS II. In MHIS I, GWU was responsible for 64.8% of the total number of claims, followed by maternal care packages (normal delivery, normal

**Table 2. Distribution of claims by type of service providers.**

| Hospital Type | MHIS-I | MHIS-II | MHIS-III |
|---|---|---|---|
| *Number of Claims and Share of Public & Private Hospitals (in %)* | | | |
| Public | 19937 (48%) | 45129 (58%) | 75997 (52%) |
| Private | 21868 (52%) | 33194 (42%) | 69215 (47%) |
| Other states* | 83 (0.1%) | 157 (0%) | 1280 (1%) |
| **Total** | 41888 (100%) | 78480 (100%) | 146492 (100%) |
| *Amount Claimed (in INR Million) and Share of Public & Private Hospitals (in %)* | | | |
| Public | 68.87 (41%) | 259.50 (50%) | 367.05 (39%) |
| Private | 95.76 (58%) | 250.45 (48%) | 538.59 (57%) |
| Other states* | 1.62 (1%) | 7.32 (1%) | 42.67 (4%) |
| **Total** | 166.25 (100%) | 517.28 (100%) | 948.31 (100%) |

* Other states indicates that care was provided outside of the state of Meghalaya

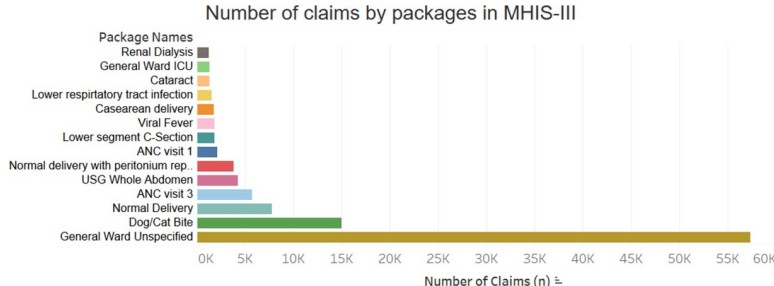

**Fig 1. Highest volume packages by number of claims for MHIS III.** Key: ICU–intensive care unit; ANC–antenatal care.

delivery with episiotomy and peritoneal repair, caesarean delivery including lower segment C-section; 14.7%) and 'General ward -ICU' (4%).

**Amount claimed.**   Fig 3 provides a summary of the packages that accrued the highest financial dispensation across MHIS III.

The package code 'General Ward Unspecified' (GWU) accounted for almost half of all spending (48%, INR 239,246,000), followed by normal deliveries (9%, INR 44,834,175) and C-section deliveries (7%, INR 35,098,065). Lower respiratory tract infection (4.1%, INR 20,648,420), intraocular lens replacement for cataract (4.5%, INR 22573237), and viral fever (4.4%, INR 21961387) are other notable areas of high spending

Fig 4 provides a summary of the packages that accrued to highest financial dispensation in MHIS II.

In MHIS II, the top three packages by volume of spending were GWU (52%, INR 195,632,000), Malaria (8%, INR 30,344,100) and normal deliveries (7%, INR 27,480,000). In MHIS I, these were GWU (47%, INR 75,023,931), 'General ward-ICU' (6%, INR 10,053,250), followed by lower segment C-section (6%, INR 9,289,500), normal delivery with episiotomy (6%, INR 8,932,750) and normal deliveries (5%, INR 8,426,875).

## Further analyses

A more in-depth analysis was undertaken for the two highest-volume claims packages, 'General Ward Unspecified (GWU)' and 'Dog/Cat bite', in order to better understand observed trends in relation to provision of care and financial spending under these packages.

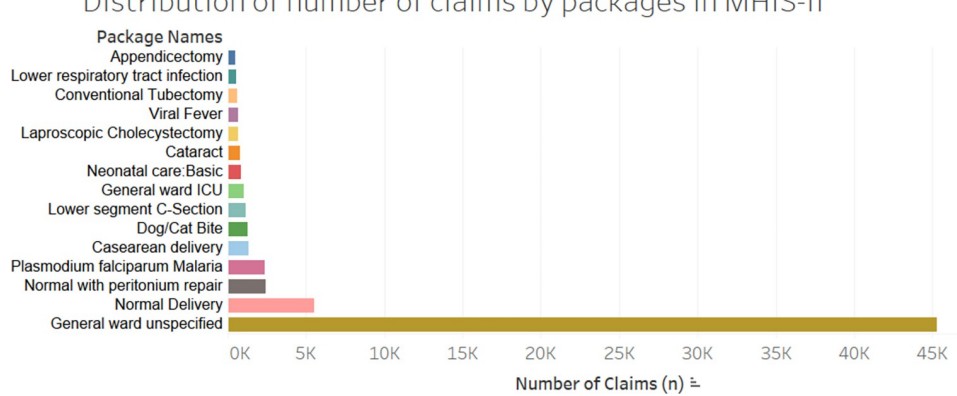

**Fig 2. Highest volume packages by number of claims for MHIS II.** Key: ICU–intensive care unit; ANC–antenatal care.

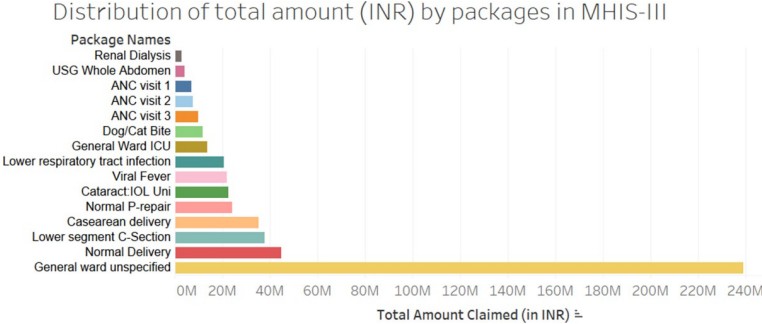

**Fig 3. Financial provision (INR, in millions) for packages claimed under MHIS III.** Key: ICU–intensive care unit; ANC–antenatal care.

**General Ward Unspecified (GWU).** The raw number of claims made for GWU doubled from MHIS I (26,892) to MHIS III (57,337), however, the number of these claims as a proportion of the total number of claims reduced from 65% to 42%. Females of 19–45 years were the highest claimants under this category in MHIS III. Analysis of manual entries of reason for claim revealed that acute gastroenteritis contributed to the highest proportion of claims under the GWU category (21%) followed by acute respiratory tract infection (13%). I. Other conditions categorized under GWU included recurrent vomiting with dehydration, typhoid and viral fever, urinary tract infection, reproductive and child health, dysentery, accelerated hypertension, auditory processing disorders, scrub typhus and cancer (site unspecified). The median length of stay in a hospital under the GWU category was 4 days across all three phases of the MHIS. Claims made for GWU were also more frequent in public as compared to private facilities (55% and 45% respectively). The average amount claimed (INR) under the GWU package in public hospitals was INR 4408 and INR 4193 in private hospitals. Average amount claimed under this category was similar to that in MHIS II (INR 4408 and INR 4193 respectively in public and private hospitals). The average amount claimed under this category in MHIS II and III is substantially higher than that of the average amount claimed under this package in the MHIS I (INR 2916 and INR 2647 respectively in public and private hospitals).

**Dog/Cat bite.** Claims made under the 'dog/cat bite' category, indicating the administration of an anti-rabies injection, contributed to the second highest volume of claims (11%) in MHIS III as compared to 1.6% of all claims in MHIS II. This package was not included in MHIS-I. A high financial provision for claims made under dog/cat bite was made in MHIS II (1.2%, INR 4386375), and this increased substantially in MHIS III (2.3%, INR 11643535). The majority of these claims were made in public sector facilities in both phases of MHIS (96% in

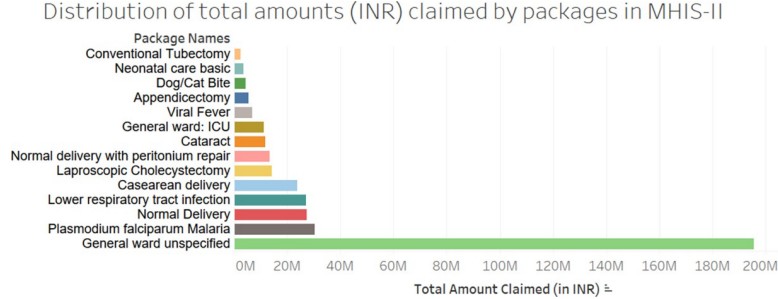

**Fig 4. Financial provision (INR, in millions) for packages claimed under MHIS III.** Key: ICU–intensive care unit; ANC–antenatal care.

MHIS III and 98% in MHIS II). When Cat/Dog bite data for MHIS-III were disaggregated by district, analysis showed that East Khasi Hills (20%) and West Garo Hills (12.5%) were the top contributing districts to these high claims in the state, however this aligns with general volume of claims disaggregated by district for all claims against population density

## Discussion

In this paper, we present a novel and comprehensive descriptive analysis of insurance claims and enrolment data from the Megha Health Insurance Scheme (MHIS) in the North East of India over a period of six years from its inception in 2013 through to 2018. This is the first analysis of the operational performance of the scheme against its UHC-oriented objectives to provide equitable access to essential healthcare for the state's population. Important insights into the provision of and spending on publicly-funded healthcare in the state of Meghalaya are presented in this paper.

We estimate that approximately 55% of the total eligible population are currently enrolled in MHIS. This is in line with the estimated 56% population-level insurance coverage published by the National Sample Survey Office of India [14]. Enrolment data analyses demonstrated an increasing pattern of enrolment over time, with the sharpest increase in number of enrolled beneficiaries between MHIS I and MHIS II. Enrolment has been highest in the age-group 19–45 years across all phases, matching demographic distribution of population by age. We observed a small decline in the enrolment of males within the 19–60 years age bracket in the transition of MHIS II to MHIS III. Upon further examination, we hypothesize two primary drivers for this. Firstly, the period upon which MHIS II was open was relatively short (1yr 4 months) as compared to MHIS I (2yrs 2 months) and III (1yr 9 months), narrowing the window within which enrolments could be processed. Secondly, enrolment kiosks for MHIS are open during usual working hours, possibly limiting those who work during office hours from easily enrolling in the scheme. Given the higher proportion of 19–60-year-old males in formal employment, as compared to women of the same age, it is perhaps unsurprising that we would observe a decline in growth of enrolment in this demographic. We also observed a decrease in enrolment growth between the different phases of the scheme in some districts as compared to others. Combined, our findings indicate that the Government of Meghalaya is on track to providing Universal Health Coverage (UHC) to the state's population with over half of all eligible beneficiaries enrolled. However, in order to reach the remaining 45% of eligible beneficiaries, enrolment campaigns will need to target those demographics for which a decline in enrolment growth has been observed. Targeted activities could include lengthening enrolment kiosk opening hours to enhance the window of opportunity for enrolment for those employed to work during core business hours, and the implementation of mobile enrolment clinics to ensure expansion of enrolment in more rural and remote districts.

Analysis of claims data by provider showed that more than half of all claims accrued to the private sector facilities across all phases of the scheme. This is highly disproportionate to facility availability, where public facilities empanelled under the scheme outnumber empanelled private facilities across the state at a rate of almost 10 to 1. However, this is a common observation across India where members of the public frequently cite perception of superior quality in the private sector as a reason for favouring private facilities over those provided by the public sector [15–17]. Almost 90% of the state's private facilities are located in the capital city Shillong. While private facilities are more likely to provide specialized care services, the highly inflated rate of service provision in private facilities in comparison to public hospitals outstrips expected demand for specialized services as indicated by state-level burden of disease [2, 18]. With 80% of the state population residing in rural areas, travel to Shillong for private facility

care is impeding the realization of UHC, which requires equitable access to high quality care. Efforts by the Government of Meghalaya to mitigate this disparity in care delivery should be bi-directional to both assess and address the potential requirements of public facilities towards strengthening capacity to deliver more specialist and high-quality care, and to work with and educate the local community regarding the accessibility and quality of care available at public facilities.

Analysis of claims data by both amount spent and by volume indicate that the package General Ward Unspecified (GWU) contributed to the highest volume of claims and the most substantial financial provision for any package of care across all three phases of the MHIS. GWU claims were made more frequently in public as compared to private facilities, and the amount claimed under this category in MHIS II and III doubled compared to that spent on the same package under MHIS I. Similar overuse of this package has been reported in Odisha previously. Where reimbursement rates are pre-determined per package by the MHIS, the GWU is unique in that there are no such pre-defined parameters for reimbursable amount. Meticulous analysis of hand-entered reasons for claims under this package revealed that the most common entries for this claim were for pre-existing packages, including typhoid, vomiting and diarrhoeal disease, and fever. When comparing amount claimed under GWU for these manual entries against the set reimbursement rate for the correct package of care, we noted that oftentimes the pre-set package rate would have been higher had the correct package been allocated. This indicates that GWU is most likely being used in administrative error due to an inability or unwillingness to locate the correct package of care. The use of the GWU package as a cover-all for conditions ranging from everything from fever to cancer could impede the state's ability to match spending to clinical need and apply adequately rigorous governance mechanisms to both clinical care provision and hospital claims management under the MHIS. Further qualitative analysis investigating healthcare provider and administrator use of the GWU package could be utilised to inform future policy targeted at enhancing the correct use of package codes and decreasing the overuse of the GWU. Information architecture could also be updated to incorporate enhanced digital accountability into the processing of certain 'red flag' packages such as the GWU, which necessitate additional data to be input before claims are processed [19].

The highly unexpected finding of claims for dog and cat bite across MHIS II and III, accounting for close to 20% of all claims, second only to GWU in MHIS III, is a cause for significant concern. This package is unique in that for a complete course of anti-rabies injections for dog and cat bite, a potentially-infected individual is required to attend the facility on five separate occasions, one for each anti-rabies injection. The care provider is permitted to submit a separate claim for the individual upon each presentation for injection [20]. State-level rabies surveillance statistics reveal low levels of animal infection and very low transmission to humans in Meghalaya [21]. However, given that anti-rabies injections are intended to be administered as post-exposure prophylaxis, the frequency of animal bite is a more important indicator of need. Local media reports suggest that cases of dog bites are indeed increasing at an alarming rate in the state, presenting a major challenge to both individuals and the health systems that provide their care [22, 23] Given the extremely high rate at which anti-rabies injections are being administered to the Meghalayan population under the MHIS, and the substantial financial provision being made for this, further investigation should be undertaken as a matter of urgency to better understand the situation and identify potential cost-effective policy solutions to reduce the societal burden of animal bites in the community.

There are a number of limitations to this analysis that should be recognised. Firstly, we provide here only top-level descriptive statistics of patterns of enrolment, volume of claims, and amount spent on claims over time. We have not triangulated this data with additional sources

such as census data on household spending, government reports of spending on health, nor clinical burden of disease. As such, we are limited to comment only on the data provided by the MHIS at this point in time. Secondly, the way in which MHIS claims data is manually entered into an electronic system, means that human errors are frequent, as is common for similar administrative data [24]. This significantly impeded our ability to collate and analyze data, and indeed numerous claims were excluded on the basis of unclear or incomplete entries. Thirdly, we have not undertaken a detailed fiscal space analysis necessary to provide a comprehensive overview of how the spending under MHIS relates to overall spending by the state of Meghalaya on health. Such an analysis would be beneficial to provide insights into how, where, and how much budgetary allocations are being made on health within the state across the various levels of care provision.

Future research on the nature and pattern of care provision under the MHIS is warranted for a more comprehensive impact evaluation into who the scheme is reaching, how care is being provided, and where taxpayer money is being spent under the scheme. Analyses of enrolment data may be triangulated against census information to more comprehensively assess enrolment of eligible beneficiaries against geography, gender, caste, religious or tribal status, and socioeconomic status. Service uptake and utilisation behaviour should be assessed against clinical need in order to investigate both the face validity of claims data in providing insights into the health of the population, and the quality of the scheme in providing care according to population need. Finally, a comprehensive assessment should be undertaken to assess the impact of care provision on health outcomes and financial risk protection, where similar analyses have indicated sub-par performance of other government-sponsored health insurance schemes across India [25–29].

Health insurance enrolment and claims data can provide an invaluable resource in which to assess operational performance of a scheme against its objectives. Here, we have tracked enrolment of eligible beneficiaries under the MHIS and trends in the care that was provided to them over a six year period. With over 55% of the eligible population currently enrolled in the MHIS, the government of Meghalaya is making headway in its ambitions to provide Universal Health Coverage to the state's population. Analysis of claims data indicates a number of areas for further data gathering and service improvement, including disproportionate use of private facilities to deliver public health services, an overuse of the general ward unspecified package of services, and an extremely high rate of anti-rabies post exposure prophylaxis provision in the state. The government of Meghalaya can use enrolment and claims data to facilitate real-time tracking of the operational performance of the scheme and use this information to make evidence-based decisions for future policy to enhance provision of healthcare under the MHIS.

## Acknowledgments

We thank the MHIS admin team for their cooperation and support throughout this study: Sonata Dkhar, Steven R Bareh, Alda. A. Pasi, Skhemjingmut Law, Stacy Passah, Natus D Ladia; Jopskhem Lyngwa and Dupphidalin L Ryngait.

We acknowledge the following researchers at the IIPHS for their support during the early phase of the study: Tiken Das, Roshan Ronghang, Yoorisa Pde, Ibaplielad Jana, and Barshana Goswami.

## Author Contributions

**Conceptualization:** Sandra Albert.

**Formal analysis:** Eliza K. Dutta, Selvaraju Venkatachalam.

**Methodology:** Sandra Albert.

**Project administration:** Eliza K. Dutta, Sandra Albert.

**Resources:** Sampath Kumar.

**Supervision:** Sampath Kumar, Sandra Albert.

**Validation:** Sampath Kumar, Selvaraju Venkatachalam, Laura E. Downey, Sandra Albert.

**Writing – original draft:** Eliza K. Dutta, Laura E. Downey.

**Writing – review & editing:** Eliza K. Dutta, Sampath Kumar, Selvaraju Venkatachalam, Laura E. Downey, Sandra Albert.

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
