## [Decision Letter · Decision Letter 0]

3 Feb 2022

PONE-D-22-01529An analysis of public insurance claims data from Meghalaya: Insights into the provision and quality of care in North East IndiaPLOS ONE

Dear Dr.Eliza Dutta ,

Thank you for submitting your manuscript to PLOS ONE. After careful consideration, we feel that it has merit but does not fully meet PLOS ONE’s publication criteria as it currently stands. Therefore, we invite you to submit a revised version of the manuscript that addresses the points raised during the review process.

ACADEMIC EDITOR: Manuscript submitted reflects upon vital procedural utilization and issues pertaining implemenation of an insurance with widespread partcipation across the state. Analytical insights for the sceintific world to decipher with further policy implication is welcome step. As mentioned by Reveiwer 1 , point 1 about terms of success and " Quality of care" in the title authors need to revisit/ re explain.Suggestion by the reveiwer 1 about  comparative estimates with NSSO latest round (75th?) is welcome suggestion and may be be addresssedSpecific comment by Reveiwer 2 about  'track and pattern'  along with utilisations stratifiaction on the pattern of Tertiary care/Secondary/pri  would be more helpful in analysing the objective in question. Reviewer 2 request for data provision in comment 5 will further help in enriching the validity of the content. Please address/revise/rebutt all the points raised during the review process in a constructive manner so that scientific methodology may further be strengthned. Editor specific.       Significant utilisations  under cat/dog bite category is a valid concern raised by reviewers. ARV treatment is usually administered on out-patient basis on specific days. Authors need to deliberate further about inclusion of such claims in the insurance scheme based on hospitalisation and expenses incurred during the hospital stay. Increased utilisations warrants thorough examination as mentioned by authors and reveiwer(2) also. Halt of the preventive tools provision as a policy measure is not justifiable/valid based on the various observations made. Please ensure that your decision is justified on PLOS ONE’s publication criteria and not, for example, on novelty or perceived impact.

Please submit your revised manuscript by % 3rd March, 2022%. If you will need more  time than this to complete your revisions, please reply to this message or contact the journal office at plosone@plos.org. Please include the following items when submitting your revised manuscript:A rebuttal letter that responds to each point raised by the academic editor and reviewer(s). You should upload this letter as a separate file labeled 'Response to Reviewers'.A marked-up copy of your manuscript that highlights changes made to the original version. You should upload this as a separate file labeled 'Revised Manuscript with Track Changes'.An unmarked version of your revised paper without tracked changes. You should upload this as a separate file labeled 'Manuscript'.

We look forward to receiving your revised manuscript.

Kind regards,

Gopal Ashish Sharma, MBBS, MD

Academic Editor

PLOS ONE

Journal Requirements:

 [This specific study did not receive any funding. RRH-HTA(IN) based at IIPHS receives operational funding from the Department of Health Research, Ministry of Health and Family Welfare, Government of India. The funders had no role in study design, data collection and analysis, decision to publish, or preparation of this manuscript.] 

Reviewers' comments:

Reviewer's Responses to Questions

**Comments to the Author**

1. Is the manuscript technically sound, and do the data support the conclusions?

Reviewer #1: Partly

Reviewer #2: Partly

2. Has the statistical analysis been performed appropriately and rigorously? 

Reviewer #1: N/A

Reviewer #2: No

3. Have the authors made all data underlying the findings in their manuscript fully available?

Reviewer #1: Yes

Reviewer #2: No

4. Is the manuscript presented in an intelligible fashion and written in standard English?

Reviewer #1: No

Reviewer #2: No

5. Review Comments to the Author

Reviewer #1: 1. Parameters for defining success of other schemes could be mentioned, (in terms of population coverage, service coverage and reduction in out of pocket expenditures is mentioned). Otherwise, making such a statement may is not useful in the article.

2. Most schemes were designed and implemented by states and even after RSBY was launched in 2008, States implemnted their own schemes in paralell. So the statement on demonstrating ability to absorb funds needs clarity.

3. RSBY only had a coverage of upto Rs.30,000 per family of 5 per annum, when all other schemes had a coverage of Rs.1.5 – 2 lakhs. Mentioning RSBY as comprehensive is incorrect. Authors may want to define comprehensives of RSBY in specific terms.

4. 'RSBY was not replaced' by PMJAY – authors may want to use more appropriate language such as RSBY was ‘succeeded by” by PMJAY.

5.Design of PMJAY was a consultative process where states were represented. line 90-91 is an incorrect statement.

PMJAY is implemented by the National Health Authority with guidelines being drawn through a consultative process with states having the flexibility to modify to local context.

6.Authors have only conducted a descriptive analysis of the claims. This should be clearly stated.

7. The analysis only pertains to population coverage (enrollment), claims analysis according to defined packages and prublic or private hospital where service was provided. Quality of care was not explored thus title should be revised to not be mis-leading.

8. The results could be substantiated by answering 'why' question either from program documents or literature on the scheme or experiences from other schemes.

9. To understand the scheme better - the details of coverage, especially service coverage should be mentioned either in the inrtoduction or as a specific section.

10. A comparision of results to NSSO Health and Morbidity data 2017-18 for Meghalaya State could be useful to analyse the scheme status - interms of insurance coverage, cost of coverage or out of pcokect expneditures and services /packages usually covered.

Reviewer #2: The paper seeks to analyse care provision, utilisation, and quality of the MHIS through analysis of the enrolment and claims data of the MHIS I to III, from 2013 to 2018 and to track patterns of enrolment of beneficiaries in the scheme over this period of time, trends in care utilisation behaviours, and patterns of care provision, The objectives of this analysis were threefold: to understand the success of the MHIS in reaching target beneficiaries for enrolment; to understand the patterns of care provision and key trends in both Government health spending and health service utilisation; and to examine how claims data could inform our understanding of the quality of care provided under the MHIS. This review has kept this as the objective to be attained by the paper.

The authors had access to enrolment and claims data should use them to answer the questions they had set for themselves and additional questions that arise from the data they have presented. Listed below are some instance where such questions remain unanswered.

1. For utilisation, the percentage of eligible population who are enrolled and the percentage of enrolled population who used the scheme ( weighted by the propensity to consume health care in the state) are good indicators. The percentage of eligible persons who were enrolled is provided for the total population. But the data presented has aspects that need to be explained, to meet the objectives of the paper. MHIS II to III shows a decline is some population groups e.g. Males, ages 19- 60 and above) and in some districts (East Jantia and Garo Hills). In order to ‘track the patterns of enrolment in the scheme’ it is important to answer whether this had implications to the reach of the scheme. [ The additional enrolment mentioned in the paper is accounted for by some demographic groups and districts ]. Authors could look at the enrolment as a percentage of eligible population in these categories and see if the dip is real or a function of demographic shift. Since they had access to claims data they could look for an explanation in the utilisation pattern of these groups in the earlier years to see if this impacted the willingness to enrol in later years. If the claims data will also demonstrate whether the fall off is linked to lack of geographic proximity, which will reduce access costs which are not reimbursed. This would have clarified a phenomenon found in their table and could have equity implications which are not explored.

2. The paper notes that larger share of claims (57%, INR 538,592,642) accrued to the 18 private hospitals empanelled under MHIS-III. The large network of public healthcare providers (159 in total) accounted for 39% (INR 367,048,292) during this period.

MHIS claims are for hospital based secondary and tertiary level treatment. So unless the private and government hospitals are of the same capacity, their utilisation may not be comparable. The reason for higher amount of claims for a few private hospitals could be that they have specialised treatment facilities that most government hospitals do not have. One way of verification would be the number of claims for private and government sector, another average amount per claim in government and private hospitals, or share of government and private hospitals in each package, as some packages are high cost (e.g. Laparoscopic Cholecystectomy). If private hospitals specialised in them they would get greater share of claim amount treating lesser number of cases. This could also be the result of a deliberate policy: government takes care of primary and secondary treatment and prefers to purchase tertiary care from private sector. Government hospitals appear to treat a greater percentage of patients with common ailments (55% to 45%) which is covered under General Ward Unspecified.

Since the data on enrolled hospitals are available it may be good to categorise the hospitals as tertiary, secondary and primary and look at the intra category differences between private and government.

3. Paper mentions that claims reimbursement is based on package rates, by which the amount reimbursed for a procedure is pre-determined. But average rates widely for the same procedures across the three schemes (e.g., Caesarean section.) Authors need to explain the discrepancy between the narrative and the table.

The paper notes (L 228)that the average claim amount for GWU in MHIS II and III is double that of the MHIS I but does not explain whether this is due to increase in package rate/per day of admission as the average length of stay is the same.

4. In discussion section the authors states that (L 250) the gender distribution of enrolment has

been equal and static over all phases of MHIS, indicating gender parity by enrolment since

the scheme’s inception. This is not borne by table 1. Male enrolment moved from 44.7 to 53.3 to 47.7. While gender distribution of health services may be skewed against females what is to be examined in this paper, is whether enrolment and utilisation follows the same pattern in other published studies on government funded health insurance schemes. Both the references cited deal with consumption of health care in general not access to government funded health insurance schemes.

5. Paper states that (L. 254) Analysis of claims data by provider showed that more than half of the claims were accrued to the private sector hospitals in all phases of the scheme. I could not find the data to support this in the paper.

6. The analysis of the reasons for overuse of GWU is not convincing. It is commonly seen that this happens due to ignorance, carelessness or unwillingness to take the effort to search for the correct package to book. It could also be that it being a catch all package it may reflect the sum of all conditions and procedures for which the scheme has not prescribed a package.

For procedures for which package rates exist, it needs to be verified the hospital would earn more money by GWU than the package. If the specified condition has a lower package rate, then booking under GWU will be deliberate and fraudulent. Authors also need to state if the practice was the same across private and government hospitals.

7. The policy recommendation to halt preventive vaccination for dog/cat bite is not justified by arguments. The rate of rabies infection does not determine the number of anti-rabies injections; the number of animal bites do. Easy availability of preventive vaccination reduces the number of infections. The paper reports that most of these injections were taken in government facilities, where the inventory would have to be run down to match the number of injections administered. With universal coverage of MHIS there is no incentive to fudge the figures unless the drug claimed but not used is being shipped outside the state. Before recommending a policy measure the issue needs to be examined in depth.

While this is the first analysis of the data and more detailed investigations may follow later, the paper has observations, which raise alternate explanations and should be resolved with the data referred to in the paper itself. Authors would do well to provide conclusions, to the questions they have posed, after examining alternate possibilities, based on the data they have quoted in the paper.

The paper has many unhappy usage of language which need to be corrected before publication.

6. PLOS authors have the option to publish the peer review history of their article (what does this mean?). If published, this will include your full peer review and any attached files.

Reviewer #1: No

Reviewer #2: No

---

## [Author Response · Author response to Decision Letter 0]

4 May 2022

Reviewer #1: 

1. Parameters for defining success of other schemes could be mentioned, (in terms of population coverage, service coverage and reduction in out of pocket expenditures is mentioned). Otherwise, making such a statement may is not useful in the article.

We thank the reviewer for this comment. We can confirm that we have now added lines detailing parameters for success of other schemes according to the three dimensions of Universal Health Coverage as defined by the WHO (i.e. population coverage, service coverage, financial coverage). Where we have referenced other schemes in both the introduction and discussion, we have now made to sure to clearly define the UHC dimensions of relevance to the scheme being discussed and how this relates to the operational performance of the MHIS against its UHC objectives. 

2. Most schemes were designed and implemented by states and even after RSBY was launched in 2008, States implemented their own schemes in parallel. So the statement on demonstrating ability to absorb funds needs clarity.

Thank you for this comment. We have amended this paragraph to provide greater clarity. We refer to progressive health reforms, highlighting successes of social insurance schemes Aarogysri and BSBY, and then move on to highlight that not all reforms were successful across the country due to an inability to absorb Government funding. We have provided two references for this, both of which report on substantial underspend in ‘high priority states’ in healthcare, due to poor infrastructure and inadequate human resource capacity. 

3. RSBY only had a coverage of upto Rs.30,000 per family of 5 per annum, when all other schemes had a coverage of Rs.1.5 – 2 lakhs. Mentioning RSBY as comprehensive is incorrect. Authors may want to define comprehensives of RSBY in specific terms.

We thank the reviewer for this comment. We agree with this comment and have now removed the word ‘comprehensive’ to rectify this oversight. 

4. 'RSBY was not replaced' by PMJAY – authors may want to use more appropriate language such as RSBY was ‘succeeded by” by PMJAY.

We thank the reviewer for this comment. We confirm that we have replaced the word ‘replaced’ with the words ‘succeeded by’ as requested.

5.Design of PMJAY was a consultative process where states were represented. line 90-91 is an incorrect statement.

We thank the reviewer for this perceptive comment. We had intended to reflect the centrally-administered nature of the scheme via the NHA, however we recognise that in fact the way the sentence was previously worded may have been misleading. We have amended the paragraph to reflect that PMJAY was designed according to a consultative process between Central and state health leaders, is implemented by the National Health Authority and nodal State health authorities, with states having the flexibility to modify to local context.The following text has been added/amended to address this comment: 

This scheme was succeeded in 2018 with the much-anticipated Ayushman Bharat Pradhan Mantri Jan Arogya Yojana (AB-PMJAY), which expanded both the range of services provided and the pool of potential beneficiaries, and is intended to provide free access to healthcare for some 500 million people in the country.(11) The scheme was designed through a consultative process between central and state Governments, and implemented by the National Health Authority and nodal State Health Authorities, though states have the flexibility to modify terms of beneficiary eligibility and the package of services available to suit local needs.

6.Authors have only conducted a descriptive analysis of the claims. This should be clearly stated.

Thank you for this comment. We have now amended the methods section of the paper in a number of ways to address this comment and strengthen the section. We have included the following line in the beginning of the methods section: A retrospective descriptive analysis was undertaken to identify patterns of enrollment, service provision and health spending in Meghalaya over the past six years (2013 - 2018). We have then divided the methods into 2 sections: Data, and analysis. In the analyses section, we have included 2 additional lines on how data was analyzed and descriptive statistics generated.

The following text has been added to the methods: 

A retrospective descriptive analysis was undertaken to identify patterns of enrolment, service provision and health spending in Meghalaya over the past six years (2013 - 2018). 

….. 

Descriptive statistics were generated for both enrolment and claims data, disaggregated across the three time periods of interest (MHIS I (May 2013 - July 2015), MHIS II (August 2015 – December 2016), and MHIS III (January 2017 – September 2018)), and by geographic region, gender, and age. As the entire population of Meghalaya is eligible for coverage under the MHIS, except for those employed by the Government and their families, who are covered under the Employees State Insurance Scheme (ESIS), total eligible population was calculated by extracting the number of households who identify as being covered by the ESIS from the State population. The number of MHIS enrollees was then divided by the total eligible population in order to generate information on proportion of eligible population currently enrolled in the scheme.

7. The analysis only pertains to population coverage (enrollment), claims analysis according to defined packages and public or private hospital where service was provided. Quality of care was not explored thus title should be revised to not be mis-leading.

Thank you for this comment. We have removed the word quality from the title and present our paper now with the following title: An analysis of public health insurance enrolment and claims data from Meghalaya: Insights into the provision of care in North East India

8. The results could be substantiated by answering 'why' question either from program documents or literature on the scheme or experiences from other schemes.

Thank you for this comment. We have now attempted to better answer the ‘why’ questions raised in interpreting our results in the discussion of the manuscript. We have done this in two primary ways: 1) citing additional literature pertaining to the operational performance of government-sponsored health insurance schemes within and outside of India, and 2) by conducting additional analyses to better understand key results. We have now added a number of new figures to the manuscript to better illustrate key findings and added sections to the discussion to provide a more thorough analysis of the implications of our findings. We hope this satisfies the reviewers request, as the changes that we have made in this respect have been substantial. 

9. To understand the scheme better - the details of coverage, especially service coverage should be mentioned either in the introduction or as a specific section.

Thank you for this comment. We have added more information to the introduction to provide further detail regarding the service coverage of the MHIS. The following text has been added: 

The MHIS was first launched in 2012 and provides access to secondary and tertiary care to all enrolled citizens of the State through a network of empanelled public and private hospitals. It was subsumed within the auspices of the AB-PMJAY in 2018, where service coverage is provided according to a list of included packages of care defined by the National Health Authority (NHA) and continues to be implemented by the MHIS office, who operate as the administering State Health Authority (SHA). The scheme offers financial protection in case of hospitalization due to medical emergencies and covers a wide range of health conditions and services free at the point of delivery. Only secondary and tertiary care is covered under the scheme, and pharmaceuticals are provided for, if administered directly to patients in hospital. 

10. A comparison of results to NSSO Health and Morbidity data 2017-18 for Meghalaya State could be useful to analyse the scheme status - interms of insurance coverage, cost of coverage or out of pocket expenditures and services /packages usually covered.

We thank you for this useful suggestion. We have now added a line to the discussion to add support to the credibility of our findings for health insurance enrolment by stating that our estimates obtained from MHIS enrolment data analysis broadly match population coverage estimates provided by the NSSO. Please note that while we recognize that reduction in out of pocket expenditure is an essential component of UHC, we have not explored this in our manuscript. We have analysed spending data only from Government-sponsored claims made to the MHIS, under which secondary care is provided free at the point of delivery. We have recommended in our discussion that a thorough investigation into whether being enrolled in the MHIS has afforded the Meghalaya community financial risk protection. The following text has been added: 

We estimate that approximately 55% of the total eligible population are currently enrolled in MHIS. This is in line with the estimated 56% population-level insurance coverage published by the National Sample Survey Office of India. 

Reviewer #2: The paper seeks to analyse care provision, utilisation, and quality of the MHIS through analysis of the enrolment and claims data of the MHIS I to III, from 2013 to 2018 and to track patterns of enrolment of beneficiaries in the scheme over this period of time, trends in care utilisation behaviours, and patterns of care provision, The objectives of this analysis were threefold: to understand the success of the MHIS in reaching target beneficiaries for enrolment; to understand the patterns of care provision and key trends in both Government health spending and health service utilisation; and to examine how claims data could inform our understanding of the quality of care provided under the MHIS. This review has kept this as the objective to be attained by the paper.

The authors had access to enrolment and claims data should use them to answer the questions they had set for themselves and additional questions that arise from the data they have presented. Listed below are some instance where such questions remain unanswered.

We thank the reviewer for this thorough and considered review. We would like the reviewer to know that we have significantly amended the manuscript on the basis of the comments received and believe that it is a much stronger manuscript because of these changes. We are grateful to the reviewer for taking the time to provide such substantial feedback. 

1. For utilisation, the percentage of eligible population who are enrolled and the percentage of enrolled population who used the scheme (weighted by the propensity to consume health care in the state) are good indicators. The percentage of eligible persons who were enrolled is provided for the total population. But the data presented has aspects that need to be explained, to meet the objectives of the paper. MHIS II to III shows a decline is some population groups e.g. Males, ages 19- 60 and above) and in some districts (East Jantia and Garo Hills). In order to ‘track the patterns of enrolment in the scheme’ it is important to answer whether this had implications to the reach of the scheme. [ The additional enrolment mentioned in the paper is accounted for by some demographic groups and districts ]. Authors could look at the enrolment as a percentage of eligible population in these categories and see if the dip is real or a function of demographic shift. Since they had access to claims data they could look for an explanation in the utilisation pattern of these groups in the earlier years to see if this impacted the willingness to enrol in later years. If the claims data will also demonstrate whether the fall off is linked to lack of geographic proximity, which will reduce access costs which are not reimbursed. This would have clarified a phenomenon found in their table and could have equity implications which are not explored.

We thank the reviewer for this detailed comment. We have made a number of changes to the manuscript the address this comment. Firstly, we have incorporated more detailed information into the methods regarding how eligibility was calculated. The following paragraph has been added to the methods: 

As the entire population of Meghalaya is eligible for coverage under the MHIS, except for those employed by the Government and their families, who are covered under the Employees State Insurance Scheme (ESIS), total eligible population was calculated by subtracting the number of households who identify as being covered by the ESIS from the State population. The number of MHIS enrollees was then divided by the total eligible population in order to generate information on proportion of eligible population currently enrolled in the scheme.

We undertook further analysis in order to understand the trends in regards to decline in enrolment growth for males 19-60 years, and those in certain districts, as highlighted by the reviewer. We believe that this is not representative of a demographic shift, but more likely the product of two interacting phenomena: 1) the MHIS II window was far smaller than that of MHIS I and III, therefore restricting the time in which the scheme was open for enrolment and 2) the enrolment kiosks were only open during working hours, prohibiting those of working age, who are predominantly males of 19-60years, from enrolling. We have added the following text to the discussion to highlight this. 

We observed a small decline in the enrolment of males within the 19-60 years age bracket in the transition of MHIS II to MHIS III. Upon further examination, we hypothesize two primary drivers for this. Firstly, the period upon which MHIS II was open was relatively short (1yr 4 months) as compared to MHIS I (2yrs 2 months) and III (1yr 8 months), narrowing the window within which enrolments could be processed. Secondly, enrolment kiosks for MHIS are open during usual working hours, prohibiting those in formal employment who work during office hours from easily enrolling in the scheme. Given the higher proportion of 19–60-year-old males in formal employment, as compared to women of the same age, it is perhaps unsurprising that we would observe a decline and stagnation in growth of enrolment in this demographic. We also observed a decrease in enrolment growth between the different phases of the scheme in some districts as compared to others. Combined, our findings indicate that the Government of Meghalaya is on track to providing Universal Health Coverage (UHC) to the state’s population with over half of all eligible beneficiaries enrolled. However, in order to reach the remaining 45% of eligible beneficiaries, enrolment campaigns will need to target those demographics for which a decline in enrolment growth has been observed. Targeted activities could include lengthening enrolment kiosk opening hours and mobile enrolment clinics to more rural and remote districts.

2. The paper notes that larger share of claims (57%, INR 538,592,642) accrued to the 18 private hospitals empanelled under MHIS-III. The large network of public healthcare providers (159 in total) accounted for 39% (INR 367,048,292) during this period.MHIS claims are for hospital based secondary and tertiary level treatment. So unless the private and government hospitals are of the same capacity, their utilisation may not be comparable. The reason for higher amount of claims for a few private hospitals could be that they have specialised treatment facilities that most government hospitals do not have. One way of verification would be the number of claims for private and government sector, another average amount per claim in government and private hospitals, or share of government and private hospitals in each package, as some packages are high cost (e.g. Laparoscopic Cholecystectomy). If private hospitals specialised in them they would get greater share of claim amount treating lesser number of cases. This could also be the result of a deliberate policy: government takes care of primary and secondary treatment and prefers to purchase tertiary care from private sector. Government hospitals appear to treat a greater percentage of patients with common ailments (55% to 45%) which is covered under General Ward Unspecified. Since the data on enrolled hospitals are available it may be good to categorise the hospitals as tertiary, secondary and primary and look at the intra category differences between private and government. 

We thank the reviewer for this comment. We recognize that it is true that the private hospitals empaneled under the MHIS are largely specialty and super-specialty hospitals, while most public facilities lack such expertise and infrastructure. However, we believe that the primary driver in private care seeking and delivery goes far beyond specialty clinical need as outlined in the Meghalayan burden of disease estimates. We believe that there are important implications for health system strengthening and universal health coverage from the disproportionate use of private hospitals for government-funded care in the State, of which 90% are located in the capital despite 80% of the population living In rural communities. We have added new references and the following text to the manuscript to address this point: 

Almost 90% of the state’s private facilities are located in the capital city Shillong. While private facilities are more likely to provide specialized care services, the highly inflated rate of service provision in private facilities in comparison to public hospitals outstrips expected demand for specialized services as indicated by state-level burden of disease.(2,19) With 80% of the state population residing in rural areas, travel to Shillong for private facility-delivered care is impeding the realization of UHC, which requires equitable access to high quality care. Efforts by the Government of Meghalaya to mitigate this disparity in care delivery should be bi-directional to both assess and address the potential requirements of public facilities towards strengthening capacity to deliver more specialist and high-quality care, and to work with and educate the local community regarding the accessibility and quality of care available at public facilities. 

3. Paper mentions that claims reimbursement is based on package rates, by which the amount reimbursed for a procedure is pre-determined. But average rates widely for the same procedures across the three schemes (e.g., Caesarean section.) Authors need to explain the discrepancy between the narrative and the table.

The paper notes (L 228) that the average claim amount for GWU in MHIS II and III is double that of the MHIS I but does not explain whether this is due to increase in package rate/per day of admission as the average length of stay is the same.

Response: 

We thank the reviewer for this comment, which we believe reflects inadequate explanation of the ‘General Ward Unspecified (GWU)’ on our behalf, rather than data discrepancy or inaccuracy. There are pre-determined package rates for all procedures carried out under MHIS, and this is true for all packages except the GWU. This is why we present total estimates of volume of care and financial dispensation for all packages in the newly added figures 1a,b, and 2a,b, and provide further detail on ‘average’ amount claimed per claim in general ward unspecified because this package is unique. We have added the following text to make this point more clearly: 

Where reimbursement rates are pre-determined per package by the MHIS, the GWU is unique in that there are no such clear parameters for reimbursable amount. It appears that this lack of clear boundaries for reimbursement under this package is contributing to its overuse. Meticulous analysis of hand-entered reasons for claims under this package revealed that the most common entries for this claim were for pre-existing packages that already exist within the MHIS, including typhoid, vomiting and diarrhoeal disease, and fever. When comparing amount claimed under GWU for these manual entries against the set reimbursement rate for the correct package of care, we noted that oftentimes the pre-set package rate would have been higher had the correct package been allocated. This indicates that GWU is most likely being used in administrative error due to an inability or unwillingness to locate the correct package of care.

4. In discussion section the authors states that (L 250) the gender distribution of enrolment has been equal and static over all phases of MHIS, indicating gender parity by enrolment since the scheme’s inception. This is not borne by table 1. Male enrolment moved from 44.7 to 53.3 to 47.7. While gender distribution of health services may be skewed against females what is to be examined in this paper, is whether enrolment and utilisation follows the same pattern in other published studies on government funded health insurance schemes. Both the references cited deal with consumption of health care in general not access to government funded health insurance schemes.

We thank the reviewer for this comment. We considered and discussed this comment in great detail as a team and decided to remove the line on gender parity in enrolment and utilization, as we felt that we did not currently have all the information necessary to substantiate this point. The primary reason for this is that we feel that further analyses is required to better understand care utilisation through a gender lens by examining claims for common conditions that affect both men and women, such as cardiovascular disease and diabetes, against population-level estimates of disease burden, to assess whether both males and females are equitably accessing care. At present, care utilisation is skewed in the direction of females under the MHIS due to the high proportion of claims being made for maternal care packages. There are well-recognised reasons for this, such as the incentivization of ASHA workers to accompany women to institutional care for their delivery. We did not have time to undertake this additional analyses at present, and hope that it may form the basis of a program of work on health system strengthening through a gender lens by a PhD student at IIPH in Meghalaya using data from MHIS I through to the more recent MHIS IV. 

5. Paper states that (L. 254) Analysis of claims data by provider showed that more than half of the claims were accrued to the private sector hospitals in all phases of the scheme. I could not find the data to support this in the paper.

Thank you for this comment. We apologise for this oversight and have added a table and accompanying text to the manuscript: 

Hospital Type MHIS-I MHIS-II MHIS-III

Number of Claims and Share of Public & Private Hospitals (in %)

Public 19937 (48%) 45129 (58%) 75997 (52%)

Private 21868 (52%) 33194 (42%) 69215 (47%)

Other States* 83 (0%) 157 (0%) 1280 (1%)

Total 41888 (100%) 78480 (100%) 146492 (100%)

Amount Claimed (in INR Million) and Share of Public & Private Hospitals (in %)

Public 68.87 (41%) 259.50 (50%) 367.05 (39%)

Private 95.76 (58%) 250.45 (48%) 538.59 (57%)

Other States* 1.62 (1%) 7.32 (1%) 42.67 (4%)

Total 166.25 (100%) 517.28 (100%) 948.31 (100%)

Analysis of the most recent claims data under MHIS III (claims made between 2017-2018) revealed that more than half of the total amount claimed during this period (57%, INR 538,592,642) accrued to the 18 private hospitals empaneled under the scheme. The large network of public healthcare providers (159 in total) accounted for 39% of claims made (INR 367,048,292) during this period. In the earlier phases of MHIS-I and II, the total amount of claims accrued to the private sector as a proportion of total amount claimed were 58% (MHIS II; INR 95,757,996) and 48% (MHIS I; INR 250,450,988) respectively. Financial provision for claims made by empanelled specialty hospitals located outside of state increased from a 1% share of total spending to a 4% share of total spending between MHIS I and MHIS III. 

6. The analysis of the reasons for overuse of GWU is not convincing. It is commonly seen that this happens due to ignorance, carelessness or unwillingness to take the effort to search for the correct package to book. It could also be that it being a catch all package it may reflect the sum of all conditions and procedures for which the scheme has not prescribed a package. For procedures for which package rates exist, it needs to be verified the hospital would earn more money by GWU than the package. If the specified condition has a lower package rate, then booking under GWU will be deliberate and fraudulent. Authors also need to state if the practice was the same across private and government hospitals. 

We thank the reviewer for this very useful comment. We also refer you to our related response to comment 3 above. We have taken this comment into great consideration, undertaken additional data-checking and analysis, and amended the manuscript accordingly. Please see the additional text that has been added to the discussion section of the manuscript below: 

Where reimbursement rates are pre-determined per package by the MHIS, the GWU is unique in that there are no such clear parameters for reimbursable amount. It appears that this lack of clear boundaries for reimbursement under this package is contributing to its overuse. Meticulous analysis of hand-entered reasons for claims under this package revealed that the most common entries for this claim were for pre-existing packages that already exist within the MHIS, including typhoid, vomiting and diarrhoeal disease, and fever. When comparing amount claimed under GWU for these manual entries against the set reimbursement rate for the correct package of care, we noted that oftentimes the pre-set package rate would have been higher had the correct package been allocated. This indicates that GWU is most likely being used in administrative error due to an inability or unwillingness to locate the correct package of care.

In regards to whether this trend was observed in both public and private facilities, we include the following text in the results section of the paper: 

Claims made for GWU were more frequent in public as compared to private facilities (55% and 45% respectively). 

7. The policy recommendation to halt preventive vaccination for dog/cat bite is not justified by arguments. The rate of rabies infection does not determine the number of anti-rabies injections; the number of animal bites do. Easy availability of preventive vaccination reduces the number of infections. The paper reports that most of these injections were taken in government facilities, where the inventory would have to be run down to match the number of injections administered. With universal coverage of MHIS there is no incentive to fudge the figures unless the drug claimed but not used is being shipped outside the state. Before recommending a policy measure the issue needs to be examined in depth.

We thank the reviewer for this insightful comment and have reflected on this point. First, we would like to clarify that it was not our intention to recommend that the government should de-fund anti-rabies injections, rather, we had intended to raise the possibility of merging the distribution of anti-rabies injections under a single program to reduce inefficiency, where at present anti-rabies injections are funded by the government in both the MHIS and the State-wide anti-rabies program. We fully appreciate that distribution of anti-rabies vaccination is a product of animal bite and not of infection, and have sought to clarify this in the manuscript. We have also added more text to reflect on the potential causes of this high volume of anti-rabies vaccinations, and potential policy measures. 

We have significantly amended our discussion of this topic and now include the following replacement text in the discussion of dog/cat bite claims in the paper: 

State-level rabies surveillance statistics reveal low levels of animal infection and very low transmission to humans in Meghalaya.(21) However, given that anti-rabies injections are intended to be administered as post-exposure prophylaxis, the frequency of animal bite is a more important indicator of need. Local media reports suggest that cases of dog bites are indeed increasing at an alarming rate in the state, presenting a major challenge to both individuals and the health systems that provide their care.(22,23). Given the extremely high rate at which anti-rabies injections are being administered to the Meghalayan population under the MHIS, and the substantial financial provision being made for this, further investigation should be undertaken as a matter of urgency to better understand the situation and identify potential cost-effective policy solutions to reduce the societal burden of animal bites on the Meghalayan community.

While this is the first analysis of the data and more detailed investigations may follow later, the paper has observations, which raise alternate explanations and should be resolved with the data referred to in the paper itself. Authors would do well to provide conclusions, to the questions they have posed, after examining alternate possibilities, based on the data they have quoted in the paper. The paper has many unhappy usage of language which need to be corrected before publication. 

Once again, we are sincerely grateful to the reviewer for their thorough and useful comments. We have made substantial edits to the paper in its entirety and feel that it is much stronger as a result. We have gone to extra lengths to seek alternate explanations of some of our key findings, and have indeed changed our conclusions in relation to the high use of the dog/cat bite package of care. Regarding the comment on “unhappy” use of language, we have made renewed efforts to address this issue and hope the paper reads better now.

---

## [Decision Letter · Decision Letter 1]

10 May 2022

An analysis of Government-sponsored health insurance enrolment and claims data from Meghalaya: Insights into the provision of health care in North East India

PONE-D-22-01529R1

Dear Dr.Eliza K Dutta ,

At the oustet, i would  like to congratulate the authors for conducting comprehensive analysis and submitting exhaustiv response to issues raised by the both reviewers. 

We’re pleased to inform you that your manuscript has been judged scientifically suitable for publication and will be formally accepted for publication once it meets all outstanding technical requirements.

Kind regards,

Gopal Ashish Sharma, MBBS, MD

Academic Editor

PLOS ONE

Additional Editor Comments (optional):

The mansucript submitted and reviewed highlights, important procedural analytical insights in the implementation, utilisation , drivers of utilisations of MHIS in the region studied. The manuscript indicates low penetartion of MHIS as only 55% of the eligible popuation is enrolled yet. The foremost component of UHC, in achieving the left out objective is maximum coverage/enrollment of those who are eligible. State authorities and policy makers need to step up, prioritize and plan further penetration of the scheme to most underprivilged and inaccessibile one.There is need to substatiate the process along with rectification of the operational issues to realize the goal of UHC in the state.  

Reviewers' comments:

Reviewer's Responses to Questions

**Comments to the Author**

1. If the authors have adequately addressed your comments raised in a previous round of review and you feel that this manuscript is now acceptable for publication, you may indicate that here to bypass the “Comments to the Author” section, enter your conflict of interest statement in the “Confidential to Editor” section, and submit your "Accept" recommendation.

Reviewer #2: All comments have been addressed

2. Is the manuscript technically sound, and do the data support the conclusions?

Reviewer #2: Yes

3. Has the statistical analysis been performed appropriately and rigorously? 

Reviewer #2: Yes

4. Have the authors made all data underlying the findings in their manuscript fully available?

Reviewer #2: Yes

5. Is the manuscript presented in an intelligible fashion and written in standard English?

Reviewer #2: Yes

6. Review Comments to the Author

Reviewer #2: The comments have been addressed satisfactorily. However, since this document was made available by the agency policy options, derived from the data would have contributed to adding value to the paper. This could be kept in mind for future publications, as promised in the paper, from the same data.

7. PLOS authors have the option to publish the peer review history of their article (what does this mean?). If published, this will include your full peer review and any attached files.

Reviewer #2: **Yes: **Rajeev Sadanandan

---

## [Editor Report · Acceptance letter]

25 May 2022

PONE-D-22-01529R1 

An analysis of Government-sponsored health insurance enrolment and claims data from Meghalaya: Insights into the provision of health care in North East India 

Dear Dr. Dutta:

I'm pleased to inform you that your manuscript has been deemed suitable for publication in PLOS ONE. Congratulations! Your manuscript is now with our production department. 

Kind regards, 

on behalf of

Dr. Gopal Ashish Sharma 

Academic Editor

PLOS ONE